# Floral hosts of leaf-cutter bees (Megachilidae) in a biodiversity hotspot revealed by pollen DNA metabarcoding of historic specimens

**Annemarie Gous[1,2], Connal D. Eardley[2], Steven D. Johnson[2], Dirk Z. H. Swanevelder[1], Sandi Willows-Munro[2]***

**1** Biotechnology Platform, Agricultural Research Council, Pretoria, South Africa, **2** School of Life Sciences, University of KwaZulu-Natal, Pietermaritzburg, South Africa

\* willows-munro@ukzn.ac.za

**Data Availability Statement:** All the data generated in the study has been uploaded to FigShare and has been made public. The DOI are as follows: : M. feline (https://doi.org/10.6084/m9.figshare.

## Abstract

South Africa is a megadiverse country with three globally recognised biodiversity hotspots within its borders. Bees in particular show high diversity and endemism in the western part of the country. Not much is currently known about the floral host preferences of indigenous bees in South Africa, with data only available from observational studies. Pollen metabarcoding provides provenance information by utilising DNA analyses instead of floral visitation and traditional microscopic identification to identify pollinator food plants, which can be time consuming and imprecise. In this study, we sampled pollen from leaf-cutter bees (Megachilidae) specimens maintained in a historic insect collection (National Collection of Insects, South Africa) that were originally collected from two florally important areas in South Africa (Succulent Karoo and Savanna) and used metabarcoding to determine pollen provenance. We also sampled pollen from leafcutter bee species with wider distributions, that extend across many different biomes, to determine if these 'generalist' species show relaxed floral host specificity in some biomes. Metabarcoding involved sequencing of the nuclear internal transcribed spacer 2 (ITS2) region. Amplicons were compared to a sequence reference database to assign taxonomic classifications to family level. Sequence reads were also clustered to OTUs based on 97% sequence similarity to estimate numbers of plant species visited. We found no significant difference in the mean number of plant taxa visited in the Succulent Karoo and Savanna regions, but the widespread group visited significantly more floral hosts. Bees from the widespread group were also characterised by a significantly different composition in pollen assemblage. The time since specimens were collected did not have an effect on the mean number of taxa visited by any of the bee species studied. This study highlights national history collections as valuable sources of temporal and spatial biodiversity data.

## Introduction

The use of next-generation sequencing (NGS) technology and DNA barcoding in high-throughput identification of plant origins from pollen samples has been on the rise in recent years [1–6]. There are many advantages to this method over traditional microscopic palynology, including increased time-efficiency, multiplexing samples to reduce costs, and the relative

12994067); M. karooensis (https://doi.org/10.
6084/m9.figshare.12994115); M. maxillosa
(https://doi.org/10.6084/m9.figshare.12994148);
M. murina (https://doi.org/10.6084/m9.figshare.
12994223); M. niveofasciata (https://doi.org/10.
6084/m9.figshare.12994238). Data for M. venusta
are available in the European Nucleotide Archive
(ENA) under project accession PRJEB14178
(http://www.ebi.ac.uk/ena/data/view/PRJEB14178).

**Funding:** The authors received no specific funding
for this work.

**Competing interests:** The authors have declared
that no competing interests exist.

ease of the process that does not require a trained palynologist [7]. Genetic analyses of pollen also allows for more accuracy, since pollen morphology is not diagnostic in all species [8]. Pollen identification through metabarcoding has been shown to be consistent in comparison to microscopy-based identifications [2, 9, 10], but there is no consensus yet regarding the consistency of pollen quantification between the two methods [2, 11].

The applications of pollen metabarcoding are extensive, and include monitoring food- and air-quality, forensic studies, and reconstructing ancient plant communities among others [7]. Metabarcoding pollen loads sampled from pollinators, as opposed to conducting lengthy field experiments, can provide insight into plant-pollinator interactions. There is still a great deal to learn about interactions between most pollinators and the plants they visit and metabarcoding provides a molecular tool for the rapid identification of pollen with mixed origin. In particular, very little is known about the floral choices of many endemic bee species in South Africa, and their interactions with plants are currently only studied through direct observations (eg. [12]) or observations combined with micoscopic examination of pollen loads (eg. [13–16]).

South Africa is exceptionally rich in plant and animal diversity, with insects accounting for the majority of animal species present within the country [17–19]. Plant and bee diversity and endemism is particularly high in the western part of the country [17, 20–22] with high levels of specialisation in mutualisms reported for this biologically important area [12]. Specialist bee and plant species are more sensitive to changes in the environment and consequently more vulnerable to declines and extinction [23, 24]. This is particularly troubling in the face of global climate change. Given the susceptibility to changes in the environment and the high levels of endemism in South Africa, there is a drive to study and better understand the interactions between bees and plants of important biodiversity hotspots.

Taxonomists, collecting bees for identification purposes, have filled insect collections with specimens from a wide range of taxa. These collections tend to be well maintained with information included with each specimen on when, where and by whom the specimen was collected [25]. Specimen labels can also include information about plant associations, but this is often lacking. Bees in these collections are not stripped of their pollen loads during their taxonomic identification and thus contain a wealth of information about the plants they visited before capture. This untapped resource can be taken advantage of by sampling pollen from insect collection specimens for metabarcoding. Recently, research showed that plant origin identifications could be made from pollen sampled from historic insect collection specimens dating back to 1914 using the internal transcribed spacer regions (ITS1 and ITS2; [26]). In this latter study, using ITS2 as a barcode, provided much better resolution for plant classification than ITS1. Unfortunately, the DNA barcode reference library for South African plants is not complete and species- and genus-level plant classification were problematic, although family-level classification was possible [26]. Nonetheless, cataloguing the number of plant OTUs visited by bees with different habitat preferences will provide valuable information.

The use of historical specimens can greatly increase the scope of questions that can be asked, as well as increase sample sizes (both temporal and spatial) available for use in studies by supplementing data obtained from field observations. In this study, pollen loads from bee specimens housed in the national insect collection in South Africa was used to determine whether the floral choice of bees in the family Megachilidae differ between the Succulent Karoo biome in the west and the Savanna region in the east of South Africa that has fewer endemics. We expected higher levels of plant and pollinator endemism to be closely linked to increased pollinator specificity and so expected megachilid species collected from the Succulent Karoo biome to visit fewer plants during foraging in comparison to species collected in the less diverse Savanna biome. In addition, we expected megachilid species that are widespread and occur in multiple biomes across South Africa to display more generalist foraging

strategies. To address these predictions, the pollen loads from megachilid specimens were metabarcoded targeting the ITS2 region of the pollen nuclear genome to reveal plant provenance and plant family affinities for each species.

## Materials and methods

### Specimen selection

Bee specimens sampled for pollen were already housed in the National Collection of Insects of the Agricultural Research Council, at the Biosystematics unit in Pretoria, South Africa. No specimens were collected for this study, which relied on the use of historical specimens. Taxonomic experts previously identified all specimens housed in the collection and the collection contains many type specimens. Six bee species from the genus *Megachile* were selected for analysis (Table 1). Two species, *Megachile karooensis* Brauns, 1912 and *Megachile murina* Friese, 1913 are endemic to the Succulent Karoo biome. The semiarid Succulent Karoo biome is distributed in the south-western portion of southern Africa and forms part of the Greater Cape Floristic Region. This biome is renowned as a globally important centre of endemism and biodiversity. The region is also well-known for high bee diversity and endemism [17, 20]. *Megachile felina* Gerstaecker, 1857 and *Megachile maxillosa* Guérin-Méneville, 1845 are restricted to the Savanna biome. This biome is the largest biome in South Africa. It is a grassland ecosystem distributed primarily in the eastern summer rainfall region of South Africa. Two widely distributed species, *Megachile niveofasciata* Friese, 1904 and *Megachile venusta* Smith, 1853 were selected as examples of generalist species.

All specimens with some pollen visible on their bodies when viewed under a dissecting microscope were selected regardless of when they were collected. One of the two widespread species selected, *Megachile venusta*, was used in a previous pollen metabarcoding study [26], where it was shown that that DNA could be successfully sequenced from a wide temporal range of samples, including pollen sampled from specimens which were collected in 1914. The data produced in this study are also used here.

Information on all bee specimens, including accession numbers, original sampling dates of bees, sampling regions and GPS coordinates (where available) is provided as supplementary information (S1–S5 Tables, and S1 Table from [26]).

A stereo dissection microscope (SteREO Discovery.V8 microscope, Carl Zeiss Microscopy GmbH, Jena, Germany) was used to view each specimen to confirm the presence of pollen, as well as to scrape pollen off abdominal scopae. Pollen samples were transferred to sterile 1.5 ml Eppendorf tubes and pollen crushed with the micropipette tip while still under magnification. The micropipette tip for each sample was left inside the tube to include any pollen that inadvertently entered the micropipette tip during scraping.

**Table 1. *Megachile* (Megachilidae) selected for study from different biomes in South Africa.** The number of specimens in the National Collection of Insects that contained pollen for sampling is also indicated, as well as the age range of specimens based on collection dates.

| Species | Biome | Number of specimens | Bee specimen collection years |
|---|---|---|---|
| *Megachile karooensis* | Succulent Karoo | 20 | 1982–1990 |
| *Megachile murina* | Succulent Karoo | 27 | 1982–1990 |
| *Megachile felina* | Savanna | 17 | 1966–1990 |
| *Megachile maxillosa* | Savanna | 32 | 1914–2003 |
| *Megachile niveofasciata* | Widespread | 10 | 1984–2000 |
| *Megachile venusta* | Widespread | 21 | 1914–2007 |

The DNA extraction protocol was optimized in a previous study [26] and used here to ensure even rupture of all pollen grains across taxa. The DNeasy® Plant Mini Kit (Qiagen, Hilden, Germany) was used without bead disruption during lysis. DNA was eluted in 50 μl of buffer EB, with reapplication of the eluate to the DNeasy Mini Spin Column for a second elution step to increase DNA yield.

The nuclear ITS2 region was selected as the barcode to be targeted for the identification of pollen's plant origins. The primers for ITS2 [27] were modified to include overhang adapters as described in the Illumina *16S Metagenomic Sequencing Library Preparation Guide* (Illumina, 2013). These overhang adapters allow the primers to be used directly in the standard Illumina indexing and adapter PCR. The primers used were ITS3F_Tag_IL 5′ **TCG TCG GCA GCG TCA GAT GTG TAT AAG AGA CAG** GCA TCG ATG AAG AAC GCA GC 3′ and ITS4R_Tag_IL 5′ **GTC TCG TGG GCT CGG AGA TGT GTA TAA GAG ACA G**TC CTC CGC TTA TTG ATA TGC 3′ (overhang adapters indicated in bold and underlined). Oligonucleotides were synthesised by Inqaba Biotechnical Industries (Pty) Ltd (Pretoria, South Africa).

Barcode amplification was achieved in reactions with a final concentration of 1 × Phusion® High Fidelity PCR Master Mix with HF Buffer (Thermo Scientific, Waltham, MA, USA), 0.5 μM of each primer, and 5 μl of DNA template. Milli- Q® $H_2O$ (Merck Millipore, KGaA, Darmstadt, Germany) was added to a final reaction volume of 50 μl. PCR conditions were as follows: 98˚C for 3 minutes, followed by 30 cycles of 98˚C for 7s, 65˚C for 30s and 72˚C for 30s, with a final step of 72˚C for 10 min. Negative controls were included in PCR-based steps. Amplified barcodes were visualised using 2% agarose gel electrophoresis.

Amplification products were purified using the Agencourt AMPure XP (Beckman Coulter, Brea, California, USA) bead purification system according to the manufacturer's protocol. The DNA concentration of approximately half of the samples was evaluated using a Qubit® 2.0 Fluorometer (Invitrogen, Life Technologies, Carlsbad, CA, USA) and a Qubit® dsDNA High Sensitivity Assay Kit (Invitrogen, Life Technologies). Samples were randomly selected for evaluation and both kits were used according to manufacturer instructions.

Pollen from five of the six bee species was sequenced using an Illumina HiSeq 2500. Pollen from *M. venusta* samples were sequenced using an Illumina MiSeq and classified in a previous study. Nextera XT (Illumina, Inc. San Diego, CA, USA) indexes were added according to the sequencing preparation protocol (Illumina, 2013) to multiplex samples. Indexed samples were pooled equimolarly and sequenced on a HiSeq 2500 sequencer (Illumina, Inc.) at the ARC's Biotechnology Platform, Pretoria, South Africa, using the HiSeq Rapid SBS Kit v2, with 2 × 250 bp paired end reads (Illumina, Inc.).

## Bioinformatics and statistical analyses

Demultiplexing of samples was done with CASAVA v1.8.2 (Illumina Part #15011196 Rev D) based on the Nextera index sequences used. Low quality bases and adapter sequences were trimmed from reads with Trimmomatic 0.35 [28] using a sliding window of eight bases with an average quality of 20 bases required per window. Trimmed reads were merged in Mac-Qiime 1.9.1–20150604 [29] using the *multiple_join_paired_ends.py* script. Trimmed sequence data is available from FigShare (see Data Availability Section).

The ITS2 database for Viridiplantae [3] was used for identification of pollen samples. Classification of sequences was performed using the Ribosomal Database Project (RDP) classifier v. 2.10.1 [30]. Analysis was performed following the bioinformatics workflow described by [30]. Reads not meeting the required 0.8 confidence level at family level after classification, and rare taxa (less than 0.1% of the total amount of reads identified as plant) were removed

prior to further analyses. Each plant family identified by RDP classifier was treated as an operational taxonomic unit (OTU). Data were handled in two different ways. Biases during DNA extraction of pollen and DNA amplification may lead to a skew in the abundances of certain taxa being identified [2, 31]. To circumvent this, plant family data were converted to absence/presence counts for each plant family for each sample in a biome, referred to as detection counts. These counts were summed across samples for each plant family to obtain plant family prevalence within a biome across all samples. Data were also analysed as read counts per sample. OTU tables were generated for read and detection counts for family abundances.

The Sickel *et al.* (2015) ITS2 database does not represent South African plant diversity adequately to make species-level classifications [26]. In order to get an estimation of angiosperm species abundance ranges in different bee species, sequence reads from each sample were clustered into OTUs using the *cluster_otus* command in usearch v8.0.1517 [32]. All reads per sample were then assigned to OTUs using *usearch_global* alignment, and taxonomy added to the OTU sequences with the *utax* command using the utax-compatible ITS2 database from [3]. During the taxonomy assignment, confidence scores were not assigned, and a raw score cut-off of 10 was chosen for a species assignment to be included. OTU tables were again created for both read counts and detection counts for species abundances, disregarding species not from angiosperm families. Rarefaction curves were drawn for plant family and species assignments for all bee species using *vegan* v. 2.3.4 [33] in R v. 3.2.4 [34]. Taxonomic assignments were checked for local family occurrence against the Plants of Southern Africa (POSA) database [35]. For the Succulent Karoo biome, the search was confined to the Western Cape, Northern Cape and Cape Floristic Region (as defined in the database), for the Savanna biome the remaining regions were selected as search criteria and for the widespread species all the South African provinces were selected. The POSA database, the vegetation map of South Africa [36], and a quarter degree grid square (QDGS) shape file were used to calculate the frequency of occurrences per family within each biome in South Africa. Spatial analyses were carried out using the *rgdal* package v.1.2–4 in R v. 3.2.4. A list of recorded flower visits (from field observation) was also generated for each bee species using available literature [37–39] and the Catalogue of Afrotropical Bees, accessed through the online Global Biodiversity Information Facility (GBIF, www.gbif.org/dataset/da38f103-4410-43d1-b716-ea6b1b92bbac [40]) for comparison.

To compare bee taxa and to assess the effect of time since collection on the mean number of taxa detectable in pollen loads, generalised linear models that incorporated a Poisson distribution and log link function were used in SPSS 23.0 (IBM Corp., Armonk, New York). Separate models were run with angiosperm species and families as response variables. To account for statistical non-independence among bees sampled from the same localities, collection locality was treated as a subject in generalised estimating equations that used an exchangeable correlation matrix. Bee species was treated as a fixed factor and time was treated as a covariate. Significance was assessed using Wald statistics, and post-hoc comparisons among means were carried out using the Dunn-Sidak procedure. Marginal (model-adjusted) means were obtained by back-transformation from the log scale, which also results in asymmetrical standard errors.

To assess whether bee species carried different assemblages of pollen, a similarity matrix for square-root transformed data on detection counts were calculated using the Bray–Curtis method and then plotted in two-dimensions with non-metric multi-dimensional scaling (NMDS) using Past 3.14 [41]. The significance of differences in pollen assemblages among bee species was assessed using ANOSIM, a non-parametric permutation procedure based on the similarity matrix underlying the ordination. Observed R-values were compared with the distribution of R-values generated by up to 10,000 random permutations of the sample labels in order to assess statistical significance.

## Results

Summary statistics of the merged reads produced in this study are provided first (Table 2). Quality and adapter trimming of samples sequenced on the HiSeq 2500 resulted in a total number of 2,947,334 high-quality, merged reads obtained across all bee species. This resulted in a mean of 28,412 merged reads per sample (median = 13,816 and SD = 43,161). Twenty-four of the 105 samples (22.6%) produced less than 1,000 reads per sample and were discarded prior to further analyses. Pollen from 21 *M. venusta* specimens [26] produced a total of 1,124,324 reads across all samples, with a mean of 53,539 reads per pollen sample (SD = 57,314). When combined with samples sequenced on the HiSeq 2500, a total of 4,071,658 reads were produced across all six *Megachile* species, after 24 failed samples were removed. This results in a mean of 40,313 (median = 24,193; SD = 48,742) reads per pollen sample. The mean number of merged reads per *Megachile* species is provided in Table 2.

ITS2 sequence classification resulted in 71.9% of total reads being confidently identified to family level across all samples. Samples with less than 1,000 reads confidently classified to family level, were discarded. Samples that passed the family-level read cut-off were all subject to species level taxonomic classification in *utax*. An additional read cut-off was not introduced at species level classification, as only reads classifying to angiosperm species were recorded. Nine samples did not pass the classifying criteria, leading to a total of thirty-three pollen samples (26.6%) across all bee species being discarded. Only reads confidently identified to family level and reads classified to angiosperm species were used to draw rarefaction curves to determine whether sequence saturation was achieved. Rarefaction curves indicated that for all bee species examined, sufficient reads were sequenced to resolve all possible plant taxa present in the pollen samples (S1 Fig).

Fig 1 show the proportions of bee specimens sampled that visited certain plant families. Notably, Asteraceae was identified in 26 samples (83.9%) in the widespread bees but only in very low proportions of samples for the other two groups (2 samples; 6.4%) in the Succulent Karoo and 2 samples (6.7%) in the Savanna. Fabaceae was also identified in a much higher proportion in the widespread samples (20 samples, 64.5%) than those in the Savanna (6 samples, 20%) or Succulent Karoo (7 samples, 22.6%).

Of the three bee groups, the widespread bee species visited the widest range of plant families with 30 families visited compared to 15 visited by the Savanna group and 18 visited by the Succulent Karoo group. After classification of reads into species OTUs, the widespread bee species again visited the most plant species, with a total of 39 different plant species identified. The Succulent Karoo bees visited a total of 25 plant species and the Savanna bees visited a total of

**Table 2. Summary statistics of merged reads (after quality and adapter trimming was performed, and subsequent merging of forward and reverse reads) of the six *Megachile* species investigated in this study.** Failed samples' reads are excluded.

| Bee species | Number of samples | Sum of reads | Mean of reads | Median of reads | Standard deviation |
|---|---|---|---|---|---|
| *Megachile felina* | 12 | 664,462 | 51,113 | 39,250 | 50,581 |
| *Megachile maxillosa* | 18 | 831,287 | 39,585 | 26,956 | 56,087 |
| *Megachile karooensis* | 15 | 362,278 | 22,642 | 9,954 | 26,406 |
| *Megachile murina* | 18 | 787,116 | 39,356 | 23,517 | 52,441 |
| *Megachile niveofasciata* | 10 | 302,191 | 30,219 | 20,800 | 21,917 |
| *Megachile venusta*[#] | 21 | 1,124,324 | 53,539 | 26,117 | 57,314 |

[#] The summary statistics represented for *Megachile venusta* are based on sequencing performed on an Illumina MiSeq, and disregards one sample which was excluded in the previous analysis (Gous *et al.*, 2019).

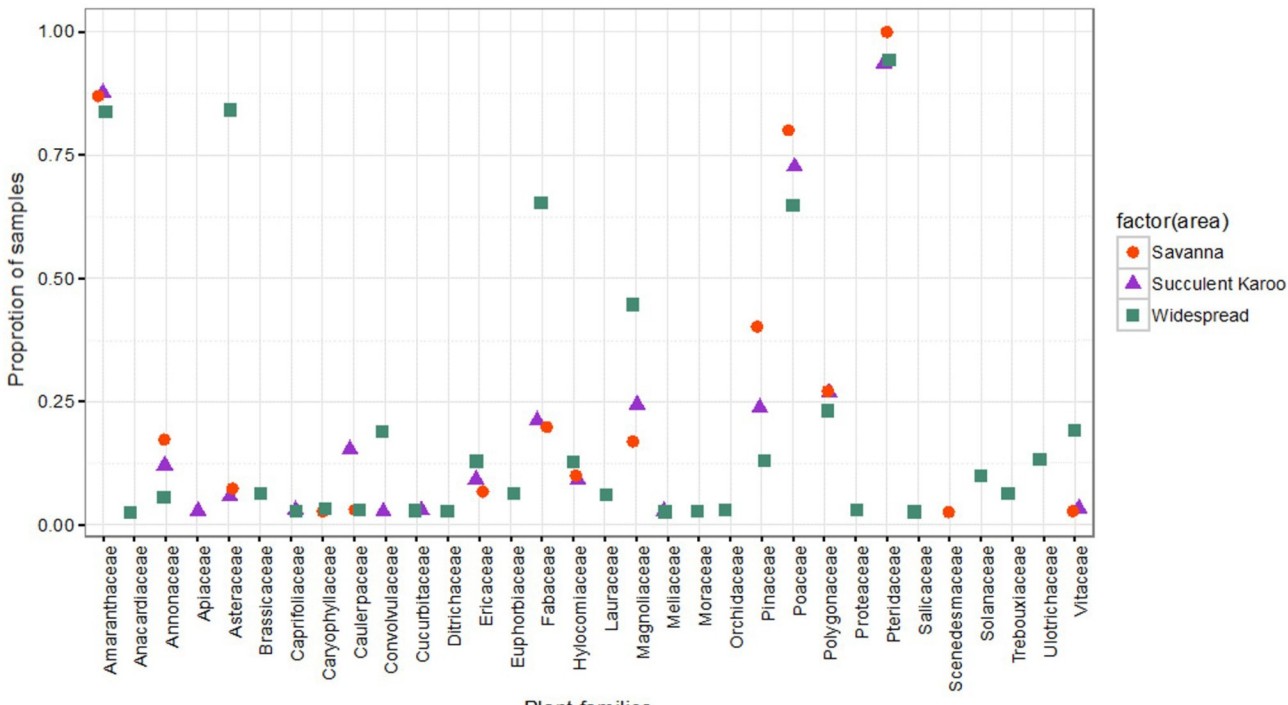

**Fig 1. Visual representation of the proportion of bee specimen samples in which each plant family was identified in each biome group.** Families from the phylum Chlorophyta were removed prior to plotting the graph. The family Pteridaceae was identified in nearly all samples in all three bee groups (Savanna, Succulent Karoo and widespread), with clear differences in the prevalence for plants belonging to Asteraceae, Fabaceae and Magnoliaceae by the widespread bee group. The absence of a group marker in a column indicates that the particular plant family was not detected in pollen loads from any bee specimens in that biome group.

16 plant species. The numbers of plant families and species visited by each bee species in each area studied can be seen in Fig 2.

The mean number of angiosperm plant families detected in pollen loads varied significantly among bee species ($\chi 2$ = 130.51, P <0.0001, Fig 3a). Similarly, the mean number of angiosperm species detected in pollen loads varied significantly among the different bee species ($\chi 2$ = 115.49, P <0.0001, Fig 3b). The highest diversity of pollen OTUs and families was recorded for the widespread species *M. niveofasciata*, but sample-based rarefaction of species OTU data indicated that the other widespread species *M. venusta* has higher expected species richness when compared to the other bee species (Fig 4). There was no significant effect of time since bee collection for either angiosperm plant families ($\chi 2$ = 0.56, P = 0.454) or the number of OTUs ($\chi 2$ = 2.28, P = 0.131).

We recorded a small, but significant difference in pollen composition (plant species recorded) between the three groups of bees as shown by NMDS (Fig 5) and ANOSIM analysis (R = 0.24, P < 0.0001). Uncorrected significance values between the different groups compared in ANOSIM (S6 Table) indicated that *M. venusta* and *M. niveofasciata* were the only two species which showed significantly different overall pollen composition.

The plant families identified in each bee group were compared to the angiosperm plant family list created from the POSA database to confirm their occurrence in South Africa. All but three angiosperm families (87.5%) in all the regions studied were found to occur in the specific region detected. Magnoliaceae was not found in any of the regions searched. Magnoliaceae species are widely cultivated which could explain their presence in the samples. Similarly, Annonaceae and Caprifoliaceae are not native to the succulent Karoo, but are widely

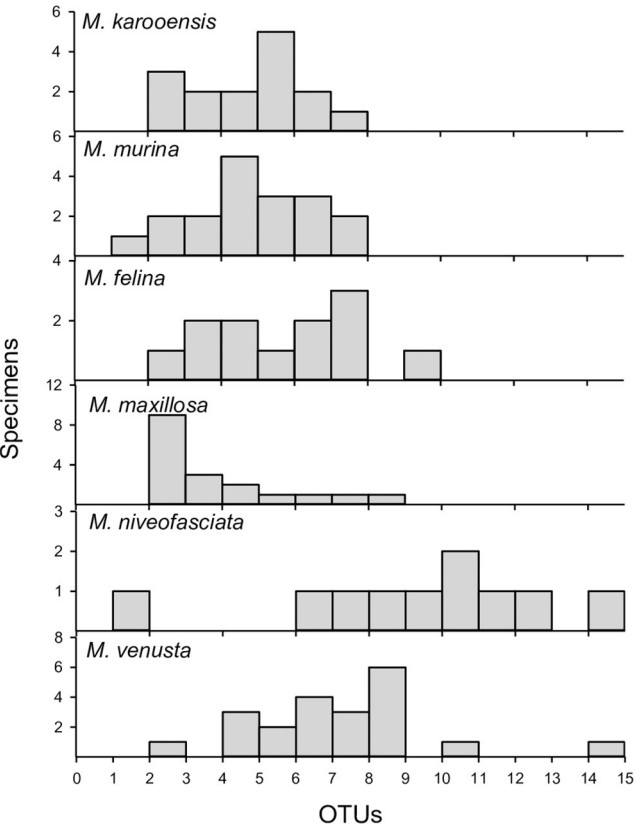

**Fig 2. Frequency distribution of plant species detected in each bee species studied.** Bees from the Succulent Karoo (*M. karooensis* and *M. murina*) and Savanna (*M. felina* and *M. maxillosa*) show similar species abundances, with the two widespread bee species (*M. niveofasciata* and *M. venusta*) both detecting up to 14 different plant species each.

cultivated. When plant families detected in pollen loads from different bee species were compared with the list of plants on which the particular bee species were observed, it was clear that Fabaceae is an important plant family for all six of the bee species (see S7 Table). Fabaceae was the only plant family with both observed flower visits and detection in pollen loads in all six bee species. Amaranthaceae was detected by NGS in all six bee species, but only recorded in flower visits for *M. karooensis* and *M. venusta*. Additional recorded flower visits corresponding to detected families in pollen loads in this study includes Asteraceae in *M. murina*, *M. felina*, *M. maxillosa*, *M. niveofasciata*, and *M. venusta*, and Solanaceae and Poaceae in *M. venusta*. The 20 plant families with the most species in each region were selected from the plant database and cross-referenced with the ITS2 sequence reference database to determine whether all these families are represented in the sequence database. Three of the top 20 most species-rich families in the Succulent Karoo, one family from the Savanna, and two widespread families were not represented at all in the ITS2 database.

## Discussion

Metabarcoding of pollen from pollinators is a recently developed molecular tool, which will greatly impact the research questions currently addressed by the fields of pollination biology and Melissopalynology (the study of pollen in honey). Here, pollen metabarcoding was used successfully on historic specimens from a natural history collection to investigate the floral

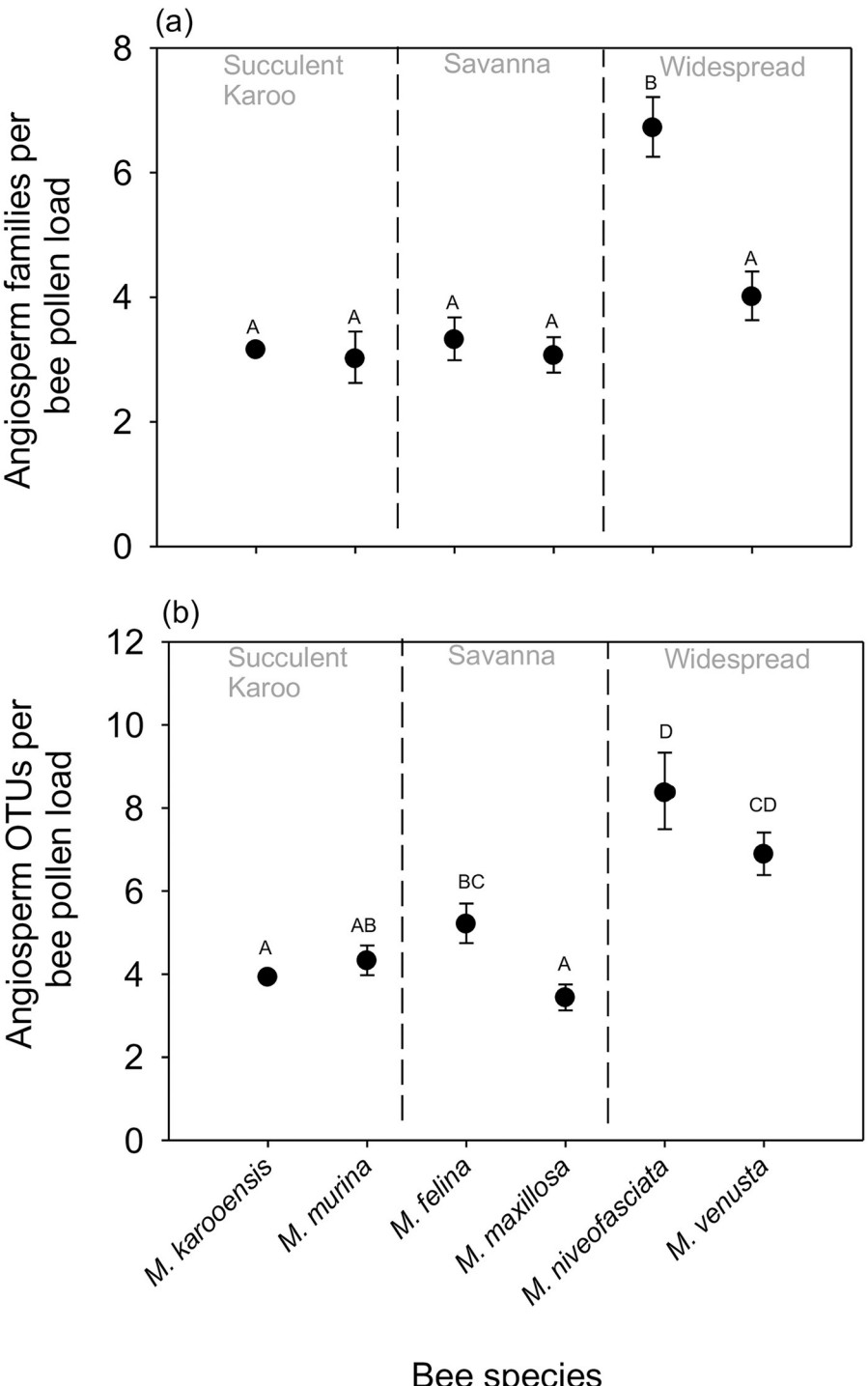

**Fig 3. Model-adjusted means for plant species OTUs detected in pollen samples in the three regions studied.** The mean number of species detected on bees from the Succulent Karoo (*M. karooensis* and *M. murina*) and the Savanna (*M. felina* and *M. maxillosa*) are lower than those on widespread bee species (*M. niveofasciata* and *M. venusta*). Means sharing a letter do not differ significantly at a 95% significance level after Sequential Dunn-Sidak post-hoc testing.

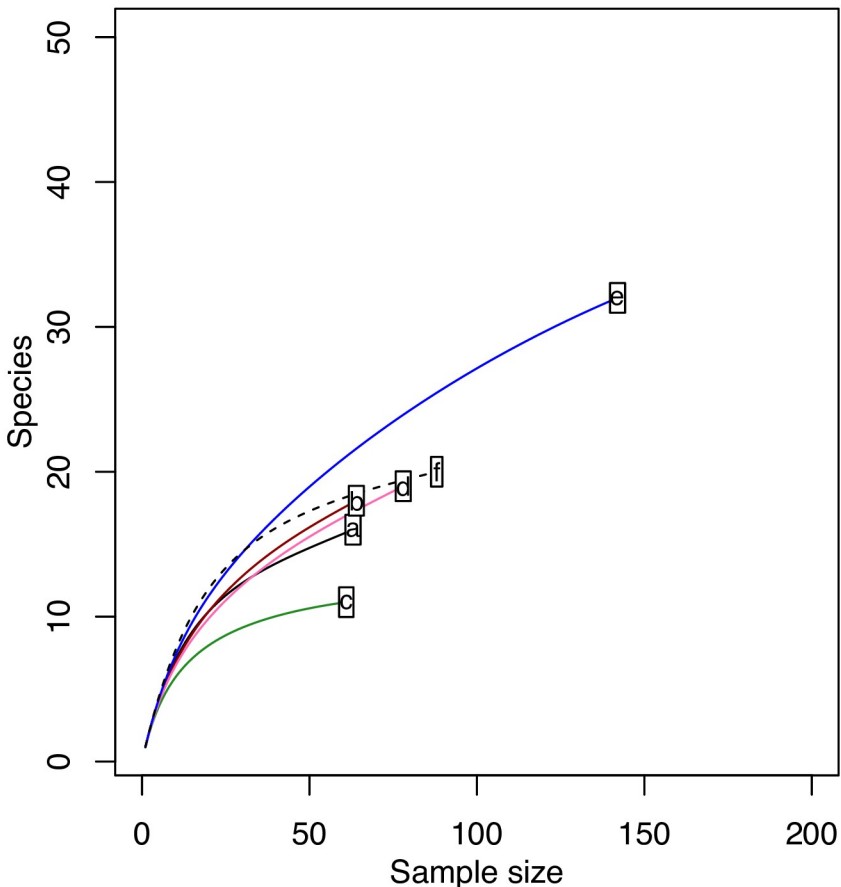

**Fig 4. Sample-based rarefaction curve to estimate plant species richness for each bee species studied.** *Megachile venusta* (e) was shown to have the highest estimated species richness. Sample labels represent bee species and are labelled as follows: (a) *M. felina* (b) *M. karooensis* (c) *M. maxillosa* (d) *M. murina* (e) *M. venusta* (f) *M. niveofasciata*.

choice differences of six species of Megachilid bees collected from different regions in South Africa. By analysing the pollen carried by these closely related species that differ in habitat preference, this study aimed to test if regions with high levels of floral endemism and species richness will also show high levels of pollinator specificity as pollinators have a known role in driving floral diversification. If bees from the more plant diverse Succulent Karoo exhibited a narrower range of floral choices than bees from the Savanna, we could expect that more plant family and species OTUs would be identified in the pollen loads of bees from the Savanna than in those from the Succulent Karoo. In contrast, the data presented here showed no significant difference in the number of plant families, or species OTUs visited by bees endemic to the Succulent Karoo and Savanna biomes. This suggests that bees in the highly plant-diverse Western region of South Africa are no more specialised than those to the less- diverse east. Our data, however, show a difference between bees endemic to just one biome with narrow habitat preference and those that forage across multiple biomes. A distinct difference was shown in the mean number of both plant family and species OTUs visited between widespread bees and those of bees endemic to either the Succulent Karoo or the Savanna. NMDS analysis supports this finding, with the widespread bee group characterised by a different pollen assemblage than that of the bee species collected in the Succulent Karoo or Savannah.

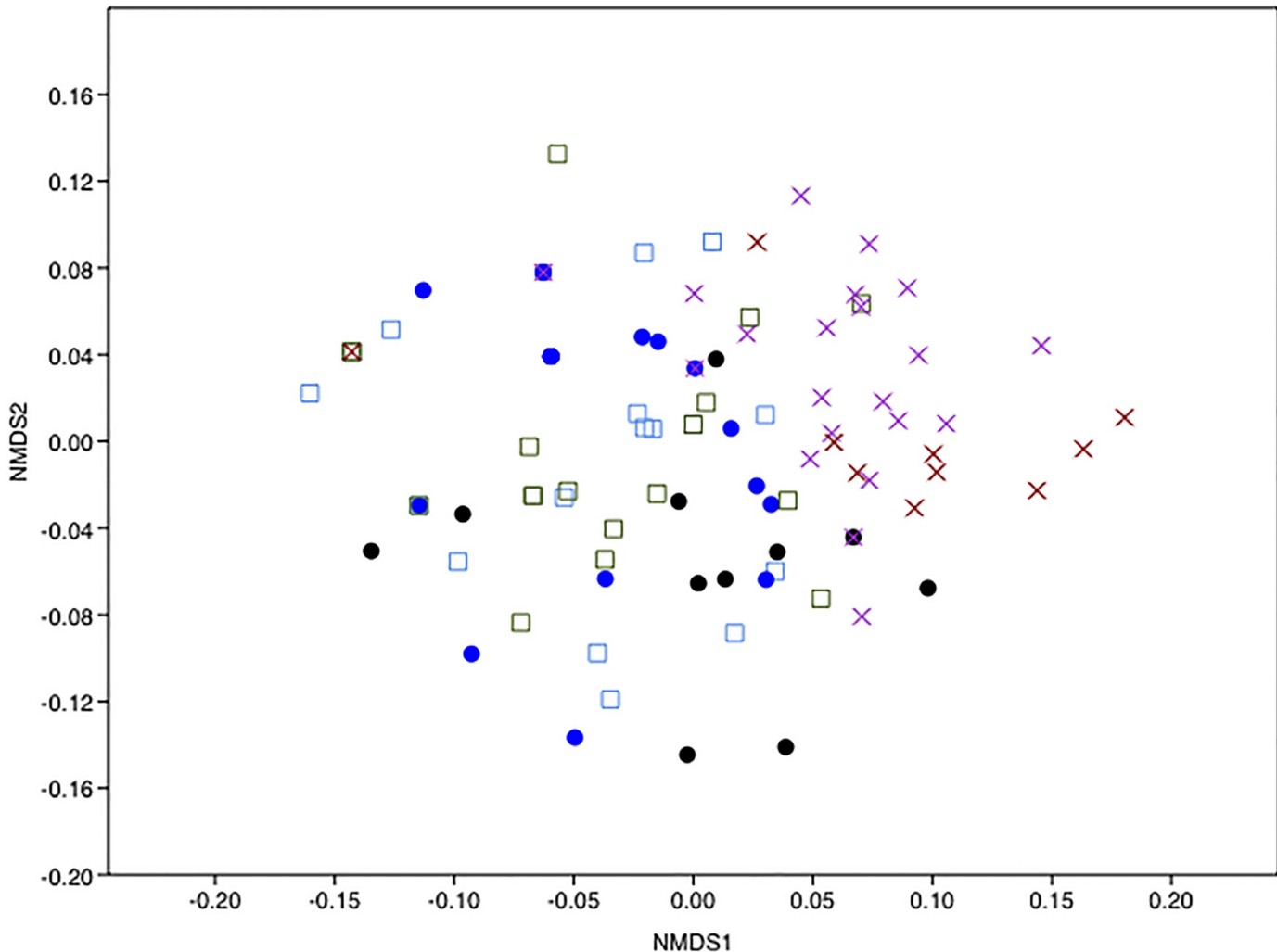

**Fig 5. Non-metric multidimensional scaling (NMDS) analysis using Bray-Curtis distances representing plant species OTUs detected on bees from the three groups.** Bee species from the same region are represented by the same shape, and different bee species colour-coded. Bees from the Succulent Karoo are represented by squares, circles represent bees from the Savanna, and crosses represent widespread species.

A striking finding from the study is that most of the plant families identified were shared among bees collected from different biomes. This suggests that although bees are endemic for a particular biome they do not predominantly visit plant species specific to that biome, but prefer to visit widespread plant taxa that occur in multiple biomes. An interesting finding was the identification of high proportions of Amaranthaceae in the overall representation of both read and detection counts in all three bee groups investigated. In the arid Western region of South Africa, the three plant families most visited by the bees are Fabaceae, Asteraceae and Aizoaceae [39]. Amaranthaceae was not in the top ten most bee-visited families, or in the list of the 15 largest plant families in the area. Only seven percent of the total visits bees made in the study were to Amaranthaceae, of which the bee family with the most visits was Megachilidae [39]. Although bees from the Megachilini tribe made no recorded visits to Amaranthaceae, it does not necessarily mean they could not be frequent visitors to this plant family. Only one of the bee species sampled for pollen in this study, *M. murina*, was mentioned in the comprehensive records of Gess & Gess (2014). The presence of this family in the Succulent Karoo is

not peculiar. Amaranthaceae is one of the top ten families in the Tankwa National Park [42] and in the top 20 families within the Extra Cape Subregion, covering 92% of the Succulent Karoo [43]. Amaranthaceae is also common in arid areas in Australasia and Eurasia. It was also found to be one of the dominant families in the Gannaveld, near Oudtshoorn [44]. To date, intensive studies of plant-bee interactions in South Africa have focused on the western part of the country, and no studies of similar magnitude have been performed in the Savanna region.

As Asteraceae is the largest plant family in South Africa and the Succulent Karoo [45], it was expected that it would be one of the most highly represented plant families identified in pollen from all biome groups, particularly in pollen from the Succulent Karoo specimens. Interestingly, in the bee subfamily Megachilinae varying rates of flower visitation has been observed between its three tribes. The Megachilini tribe visits Fabaceae flowers markedly more than the other two tribes, and the reverse is true for their visits to Asteraceae. A low flower visitation rate of only 20% was observed in the Megachilini for Asteraceae [39]. Our results showed that Fabaceae was present in the top five most visited families for all three groups but Asteraceae was only a highly represented family in pollen from widespread bees.

Grasses (Poaceae) were another highly represented family in all three studied groups. There have been reports that bees also pollinate grasses [46–50]. The Poaceae family is very highly represented in the Savanna and is one of the 15 largest families in the Succulent Karoo [39] and it is thus plausible that these *Megachile* species could collect pollen from this family.

Pteridaceae was identified in almost all pollen samples and in high proportions as well. The legitimacy of this identification is under contention as it is possible that the entries in the underlying reference sequence database originating from NCBI were problematic [26]. As Pteridaceae is a family that has 7,643 species across South Africa and 331 species occur in the Succulent Karoo and 2,177 in the Savanna [35], it is quite plausible that airborne fern spores are included in pollen loads. Also, the presence of fern spores in pollen metabarcoding studies of honey [51, 52] confirm the possibility of bees foraging on spores, even though its presence was detected in less samples and in lower frequencies.

This study illustrated once again the utility of historic insect specimens for pollen sampling in order to answer questions about floral host selection. Importantly, time since bee collection did not have a significant effect on the mean ranges of plant taxa observed, which indicates that pollen DNA was sufficiently preserved within the insect collection for analysis. Pollen loads from similarly stored museum specimens were analysed to investigate the effects of different factors on bee species decline in the Netherlands [53]. They found a significant effect for time, where the size of floral ranges observed in pollen loads from bees prior to 1950 played a key role in bee population trends.

There are limitations in the present study that need to be taken into account when considering the results presented. First, the results obtained were contingent upon the bee species chosen for study. All six of the species chosen were within the same family (Megachilidae), subfamily (Megachilinae) and tribe (Megachilini, [37, 38]). Choosing bee specimens from different subfamilies or tribes within the same family might produce different results. Second, not all specimens for the species of interest in the collection necessarily contained pollen for sampling, which means all specimens with pollen loads were selected for the study. In this study, it meant that some original sampling dates and localities were overrepresented in some species, in particular in *M. karooensis* and *M. murina*. In the widespread bee *M. niveofasciata*, specimens with pollen loads were coincidentally all caught in the western region of South Africa, whereas *M. venusta* had a broader sampling spread across the country. A limited spread of data hampers the ability to infer on floral choice differences on a species-wide scale. A uniform

spread of sampling sites across the regions studied would have been ideal, but could not be obtained here due to the limiting factor of pollen availability on historic specimens.

## Conclusion

In summary, this study showed that pollen metabarcoding can be used to answer questions about floral choice in bees from three different geographic areas. It was possible to show that bees from the Succulent Karoo and Savanna do not differ significantly in the number of plant families or species that they visit, and that widespread bees tend to visit more plant taxa than bees restricted to a single biome. Clustering sequence reads into species OTUs allowed floral choice ranges to be investigated in more detail than if only family level classifications were considered. A comprehensive DNA reference library for all flowering plants in South Africa would greatly improve the conclusions drawn from this and future metabarcoding studies. Using pollen metabarcoding can be very useful in understanding interactions between threatened pollinators and plants in highly diverse regions of the world such as South Africa and will aid conservationists in their efforts to protect these biodiversity hotspots.

One of the reasons for undertaking this study was to understand better whether climate change will have a greater effect on endemic species with narrower distributions and pollination specificity. Although bee species endemic to a small biome, such as the Succulent Karoo, do not differ much from those endemic to a large biomes, the Savannah, both are less polylectic than wide-spread species. This topic needs further investigation, concordant patterns across different pollinators would provide the best evidence for climate-driven shifts in pollinator services.

## Supporting information

**S1 Table. The national insect collection, ARC, South Africa collection information of** *Megachile venusta* **bee specimens from which pollen was collected for this study.** Collection information, such as the date, province, GPS coordinates and nearest town are given for each sample, where available. The new province and town designations are given in brackets where they have been changed.
(DOCX)

**S2 Table. Collection information from the national insect collection, ARC, South Africa, of** *Megachile murina* **bee specimens from which pollen was collected for the Succulent Karoo group in this study.** Collection information, such as the date, province, GPS coordinates and nearest town are given for each sample, where available. The new province and town designations are given in brackets where they have been changed.
(DOCX)

**S3 Table. The national insect collection, ARC, South Africa collection information of** *Megachile maxillosa* **bee specimens from which pollen was collected for the widespread group in this study.** Available information regarding the specimen collection, such as date, province, GPS coordinates and collection locality are given for each sample.
(DOCX)

**S4 Table. Collection information from the national insect collection, ARC, South Africa, of** *Megachile felina* **bee specimens from which pollen was collected for the Savanna group in this study.** Available information regarding the specimen collection, such as date, province, GPS coordinates and collection locality are given for each sample.
(DOCX)

**S5 Table. Collection information from the national insect collection, ARC, South Africa, of *Megachile niveofasciata* bee specimens from which pollen was collected for the widespread group in this study.** Available information regarding the specimen collection, such as date, province, GPS coordinates and collection locality are given for each sample.
(DOCX)

**S6 Table. Uncorrected significance (p) values between the different groups compared in ANOSIM.** Groups were set as bee species. The only two bee species differing significantly from others were the two widespread species, *M. venusta* and *M. niveofasciata*.
(DOCX)

**S7 Table. A list of plant species and families on which visits from the six studied bee species have been recorded.** References from which data were sourced are given, as well as any additional information available.
(DOCX)

**S1 Fig. Rarefaction curves for each bee species to determine whether sequence saturation has been reached with the amount of reads sequenced per pollen sample.** The lines for all samples collected from specimens from all six of the bee species flattened when 1,000 sequence reads were reached, indicating that all possible plant families represented in the pollen samples have been identified. a) and b) represents bee species occurring in the Savanna biome, c) and d) represents bee species occurring exclusively in the Succulent Karoo biome, and e) and f) represents bee species occurring all over South Africa (cosmopolitan group).
(DOCX)

## Acknowledgments

Charles Hefer and Genevieve Thompson for their assistance with bioinformatics.

## Author Contributions

**Conceptualization:** Annemarie Gous, Connal D. Eardley, Dirk Z. H. Swanevelder, Sandi Willows-Munro.

**Data curation:** Annemarie Gous.

**Formal analysis:** Annemarie Gous, Steven D. Johnson, Sandi Willows-Munro.

**Supervision:** Connal D. Eardley, Dirk Z. H. Swanevelder, Sandi Willows-Munro.

**Writing – original draft:** Annemarie Gous, Sandi Willows-Munro.

**Writing – review & editing:** Connal D. Eardley, Steven D. Johnson, Dirk Z. H. Swanevelder.

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
