## [Decision Letter · Decision Letter 0]

11 Aug 2020

PONE-D-20-18859

Floral hosts of leaf-cutter bees (Megachilidae) in a biodiversity hotspot revealed by pollen DNA metabarcoding of historic specimens

PLOS ONE

Dear Dr. Willows-Munro,

Thank you for submitting your manuscript to PLOS ONE. After careful consideration, we feel that it has merit but does not fully meet PLOS ONE’s publication criteria as it currently stands. Therefore, we invite you to submit a revised version of the manuscript that addresses the points raised during the review process.

We look forward to receiving your revised manuscript.

Kind regards,

Guy Smagghe, PhD

Academic Editor

PLOS ONE

Journal Requirements:

3. Please include a copy of Table 3 which you refer to in your text on page 12.

Reviewers' comments:

Reviewer's Responses to Questions

**Comments to the Author**

1. Is the manuscript technically sound, and do the data support the conclusions?

Reviewer #1: Yes

Reviewer #2: Yes

2. Has the statistical analysis been performed appropriately and rigorously? 

Reviewer #1: Yes

Reviewer #2: Yes

3. Have the authors made all data underlying the findings in their manuscript fully available?

Reviewer #1: Yes

Reviewer #2: Yes

4. Is the manuscript presented in an intelligible fashion and written in standard English?

Reviewer #1: Yes

Reviewer #2: Yes

5. Review Comments to the Author

Reviewer #1: Gous et al. report on the use of pollen barcoding on samples from a historic museum collection to test differences in floral host range of different Megachile spp. More specifically they test if Megachile spp. with a widespread geographical range (in South Africa) are more generalist compared to species with a more confined geographical range. The manuscript is well written and technically sound, yet I do mis the novelty and clear message of the manuscript. They state several times that the study shows that historic insect specimens can be used, yet they showed this already in their previous article with even older samples. The study indeed shows that host range can be deduces using historical samples, yet this is not really ‘novel’. This being said the authors pose an interesting hypothesis, with regard to pollinator specificity, which I think is worth further investigation using a wider range of species and a higher number of samples (keeping in mind the limitation of historic samples) to make a stronger claim.

Reviewer #2: This article is very relevant in the context of biodiversity conservation, and as stated in the conclusion, can give more understanding of possible impact of climate change on endemism. This 'search to understanding' was lacking in the discussion of the paper; the presence of plant families in the pollen loads was only explained through comparison with observation data or other literature, but not in context of ecology and possible effects on shifts in plant-bee interactions due to climate change. But to answer those type of questions, a lot more data is needed, on many different bee species. Also, historical data will not give much explanation to current effects of climate change .

The choice to work with only Megachilidae and 2 species per group (biome or widespread), is understandable but does not give opportunity to answer any questions more than 'Is metabarcoding from historic specimens possible and how does it compare to observational data'. The sentence on line 435-437 is thus not suitable and also ambiguous.

The hypothesis on line 346- 351 (and 96) is not very clear. The link between high levels of floral endemism and species richness should be more grounded.

On line 346, a first hypothesis is explained, but the second hypothesis is not stated anywhere.

Some other questions I had were:

- 25: what are the three globally recognised biodiversity hotspots and are they used/relevant to this study?

- 75: what is meant by susceptibility?

- 95: why the choice of only looking at Megachilidae, and if relevation explanation; add in text

- 110: how and why were these 6 species chosen?

- 168: why the use of HiSeq when MiSeq was used in a previous study? Will this not affect species outcome because of different sequencing depths?

- 369 until 387:on what is this knowledge about tribes based? the 6 species studied were all from the same tribe?

6. PLOS authors have the option to publish the peer review history of their article (what does this mean?). If published, this will include your full peer review and any attached files.

Reviewer #1: No

Reviewer #2: **Yes: **Tina Tuerlings

---

## [Author Response · Author response to Decision Letter 0]

28 Sep 2020

Dear Prof Smagghe (Academic Editor) and PLOS ONE editorial team,

Thank you for allowing us to resubmit a revised version of our manuscript. We have carefully gone through the issues raised by the reviewers and have addressed each below. 

All the data generated in the study has been uploaded to FigShare and has been made public. The DOI are as follows: : M. feline (https://doi.org/10.6084/m9.figshare.12994067); M. karooensis (https://doi.org/10.6084/m9.figshare.12994115); M. maxillosa (https://doi.org/10.6084/m9.figshare.12994148); M. murina (https://doi.org/10.6084/m9.figshare.12994223); M. niveofasciata (https://doi.org/10.6084/m9.figshare.12994238). Data for M. venusta are available in the European Nucleotide Archive (ENA) under project accession PRJEB14178 (http://www.ebi.ac.uk/ena/data/view/PRJEB14178). Please can the “Data Availability Statement” be updated with this information.

We hope that this manuscript now meets the standards required for publication in PLOS ONE. If you require any additional information please do not hesitate to contact me using the contact details provided during submission.

kind regards

Dr Willows-Munro (corresponding author on behalf of all authors)

---

## [Decision Letter · Decision Letter 1]

13 Oct 2020

PONE-D-20-18859R1

Floral hosts of leaf-cutter bees (Megachilidae) in a biodiversity hotspot revealed by pollen DNA metabarcoding of historic specimens

PLOS ONE

Dear Dr. Willows-Munro,

Thank you for submitting your manuscript to PLOS ONE. After careful consideration, we feel that it has merit but does not fully meet PLOS ONE’s publication criteria as it currently stands. Therefore, we invite you to submit a revised version of the manuscript that addresses the points raised during the review process.

We look forward to receiving your revised manuscript.

Kind regards,

Guy Smagghe, PhD

Academic Editor

PLOS ONE

Reviewers' comments:

Reviewer's Responses to Questions

**Comments to the Author**

1. If the authors have adequately addressed your comments raised in a previous round of review and you feel that this manuscript is now acceptable for publication, you may indicate that here to bypass the “Comments to the Author” section, enter your conflict of interest statement in the “Confidential to Editor” section, and submit your "Accept" recommendation.

Reviewer #1: (No Response)

Reviewer #2: All comments have been addressed

2. Is the manuscript technically sound, and do the data support the conclusions?

Reviewer #1: Yes

Reviewer #2: Yes

3. Has the statistical analysis been performed appropriately and rigorously? 

Reviewer #1: Yes

Reviewer #2: Yes

4. Have the authors made all data underlying the findings in their manuscript fully available?

Reviewer #1: Yes

Reviewer #2: Yes

5. Is the manuscript presented in an intelligible fashion and written in standard English?

Reviewer #1: Yes

Reviewer #2: Yes

6. Review Comments to the Author

Reviewer #1: As stated in my previous comments I miss the clear message of the manuscript. The authors claim to have "thoroughly" revised the manuscript yet deleting/adding a few words is not a thorough revision of the manuscript, furthermore the revisions do not add to a clearer message, which is still lacking. I still believe the original hypothesis put forward by the authors is very interesting, yet they need more samples to prove this.

Reviewer #2: The author has adequately answered the comments and adjusted the article; for this reason, I have no more comments other than my approval and congratulations for this valuable contribution to important biodiversity data.

7. PLOS authors have the option to publish the peer review history of their article (what does this mean?). If published, this will include your full peer review and any attached files.

Reviewer #1: No

Reviewer #2: **Yes: **Tina Tuerlings

---

## [Author Response · Author response to Decision Letter 1]

30 Oct 2020

Thank you for allowing us to respond to the comment raised by Reviewer 1. In this study we use data from multiple individuals belonging to six species of Megachile bees, which differ in habitat specificity. We feel that the data collected is appropriate for the hypothesis tested and our conclusions are supported by the data. We do not feel that adding additional data to the study will significantly change the conclusions. We received a very favourable review from Reviewer 2, who noted that this is a ”valuable contribution to important biodiversity data”. The data presented in this study was generated during a PhD. The student has completed her studies and there is no opportunity to generate more data. We are thus unable to add more samples to our study, but feel that our manuscript is strong enough for publication in PLOS ONE given the very favourable review we received from Reviewer 2. We have made no changes to the manuscript, but resubmit “as is” for consideration. 

We hope that this manuscript now meets the standards required for publication in PLOS ONE. If you require any additional information please do not hesitate to contact me using the contact details provided during submission.

---

## [Decision Letter · Decision Letter 2]

21 Dec 2020

Floral hosts of leaf-cutter bees (Megachilidae) in a biodiversity hotspot revealed by pollen DNA metabarcoding of historic specimens

PONE-D-20-18859R2

Dear Dr. Willows-Munro,

We’re pleased to inform you that your manuscript has been judged scientifically suitable for publication and will be formally accepted for publication once it meets all outstanding technical requirements.

Kind regards,

Guy Smagghe, PhD

Academic Editor

PLOS ONE

Additional Editor Comments (optional):

Reviewers' comments:

Reviewer's Responses to Questions

**Comments to the Author**

1. If the authors have adequately addressed your comments raised in a previous round of review and you feel that this manuscript is now acceptable for publication, you may indicate that here to bypass the “Comments to the Author” section, enter your conflict of interest statement in the “Confidential to Editor” section, and submit your "Accept" recommendation.

Reviewer #2: All comments have been addressed

2. Is the manuscript technically sound, and do the data support the conclusions?

Reviewer #2: Yes

3. Has the statistical analysis been performed appropriately and rigorously? 

Reviewer #2: Yes

4. Have the authors made all data underlying the findings in their manuscript fully available?

Reviewer #2: Yes

5. Is the manuscript presented in an intelligible fashion and written in standard English?

Reviewer #2: Yes

6. Review Comments to the Author

Reviewer #2: (No Response)

7. PLOS authors have the option to publish the peer review history of their article (what does this mean?). If published, this will include your full peer review and any attached files.

Reviewer #2: **Yes: **Tina Tuerlings

---

## [Editor Report · Acceptance letter]

4 Jan 2021

PONE-D-20-18859R2 

Floral hosts of leaf-cutter bees (Megachilidae) in a biodiversity hotspot revealed by pollen DNA metabarcoding of historic specimens 

Dear Dr. Willows-Munro:

I'm pleased to inform you that your manuscript has been deemed suitable for publication in PLOS ONE. Congratulations! Your manuscript is now with our production department. 

Kind regards, 

on behalf of

Prof. Guy Smagghe 

Academic Editor

PLOS ONE